# Key Pathways and Genes of *Arabidopsis thaliana* and *Arabidopsis halleri* Roots under Cadmium Stress Responses: Differences and Similarities

**DOI:** 10.3390/plants12091793

**Published:** 2023-04-27

**Authors:** Gabriella Sferra, Daniele Fantozzi, Gabriella Stefania Scippa, Dalila Trupiano

**Affiliations:** Department of Biosciences and Territory, University of Molise, 86090 Pesche, Italy

**Keywords:** heavy metal stress, cadmium stress, phytoremediation, root biology, *Arabidopsis halleri*, *Arabidopsis thaliana*

## Abstract

Cadmium (Cd) is among the world’s major health concerns, as it renders soils unsuitable and unsafe for food and feed production. Phytoremediation has the potential to remediate Cd-polluted soils, but efforts are still needed to develop a deep understanding of the processes underlying it. In this study, we performed a comprehensive analysis of the root response to Cd stress in *A. thaliana*, which can phytostabilize Cd, and in *A. halleri*, which is a Cd hyperaccumulator. Suitable RNA-seq data were analyzed by WGCNA to identify modules of co-expressed genes specifically associated with Cd presence. The results evidenced that the genes of the hyperaccumulator *A. halleri* mostly associated with the Cd presence are finely regulated (up- and downregulated) and related to a general response to chemical and other stimuli. Additionally, in the case of *A. thaliana*, which can phytostabilize metals, the genes upregulated during Cd stress are related to a general response to chemical and other stimuli, while downregulated genes are associated with functions which, affecting root growth and development, determine a deep modification of the organ both at the cellular and physiological levels. Furthermore, key genes of the Cd-associated modules were identified and confirmed by differentially expressed gene (DEG) detection and external knowledge. Together, key functions and genes shed light on differences and similarities among the strategies that the plants use to cope with Cd and may be considered as possible targets for future research.

## 1. Introduction

Cadmium (Cd) is a non-essential heavy metal and one of the most widespread pollutants in both terrestrial and marine environments with largely demonstrations about its high cytotoxicity toward all organisms [1]. Plants can remediate Cd-metal polluted soils by diverse strategies generally termed phytoremediation, which are promising, clean, and easy to apply [2]. To address phytoremediation, plants may act on the heavy metals through different schemes such as uptake (phytoextraction) or stabilization in the root system (phytostabilization) [3]. Among all, phytostabilization is of particular interest, not only for its feasibility and effectiveness but also for its being peculiar to a wide range of plants which include commercial species and the model *Arabidopsis thaliana* [4]. In *A. thaliana,* the obvious effect of Cd toxicity is a reduction of plant growth related to the inhibition of photosynthesis, respiration, and nitrogen metabolism, as well as to a reduction in water and nutrient uptake [5]. The root system, as the first organ-sensing soil heavy metal, is strongly affected by Cd, which in *A. thaliana* inhibits primary root growth via affecting the root apical meristem (RAM) stem cell niche, while lateral root formation is somehow stimulated due to changes in root radial patterning [1]. The most recent findings evidenced that many signaling molecules and plant growth regulators are involved in Cd sensing and downstream responses, which encompass modifications at both transcriptional and post-translation levels [6,7,8,9]. However, the relationship between all these factors remains unclear, differing with species, plant organs, heavy metal concentrations, and treatment durations (reviewed in [8]). Recently, *Arabidopsis halleri* gained attention for its constitutivee tolerance to zinc (Zn) and Cd and for its close phylogenetic relationship to the model plant *A. thaliana*; thus, it is now emerging as a promising model to study heavy metal hyperaccumulation traits [10,11,12]. Despite this evolutionary closeness between *A. halleri* and *A. thaliana*, the main mechanisms for tolerating metal stress are specific to one or the other species and therefore may be based on some different and particular biological processes, or the activation of them according to species-specific patterns.

*A. halleri*, as a metal(loid) hyperaccumulator [13], is attracting the research toward the use of this trait in the remediation of metal-polluted soils [10,14]. Different mechanisms underlying Cd and Zn tolerance and accumulation were found in the natural *A. halleri* population, highlighting contrasting Cd/Zn accumulation capacities and differences in the ability to adjust micronutrient homeostasis and flavonol content upon high Zn or Cd exposure [15,16]. Comparative transcriptomic studies of *A. halleri* and *A. thaliana* also identified ten genes candidates in metal homeostasis with putative roles in metal hyperaccumulation or metal hypertolerance, which encode various transmembrane transporters of metal cations and isoforms of a metal chelator biosynthetic enzyme [17,18,19]. Specifically, hypertolerance in *A. halleri* relies on the activity of genes involved in metal uptake, xylem loading, root-to-shoot translocation, and metal sequestration in the leaf vacuole or metal accumulation in the veins [15,18,20,21,22,23]. Transcript levels of Heavy Metal ATPase 4 (HMA4), which encodes a plasma membrane P1B-type metal ATPase mediating cellular export for Zn and Cd xylem loading in the root, was substantially higher in *A. halleri* than in *A. thaliana*, as was also observed for several other candidate genes such as zinc-regulated transporter and iron-regulated transported (IRT)-like proteins (ZIPs). Additionally, the cell wall has been described in *A. halleri* to play a role in the adaptation to metalliferous soils [15,16] with several genes, being involved in its response to Cd stress, interacting among each other and with polysaccharides to structure the cell wall [24,25].

However, to our knowledge, key connectors/biomarkers specifically related to *A. thaliana* phytostabilization or *A. halleri* hyperaccumulation abilities in response to Cd stress remain unclear. Thus, further insights are required mainly concerning the genetic network and functional relationships among genes/pathways involved in root growth and development under Cd stress. In particular, revealing of how proteins cooperate and participate in complexes or pathways to accomplish vital tasks is pivotal, especially for distinguishing functions and peculiarities which underlie a specific trait such as phytostabilization or hyperaccumulation. The in wet approaches are common practices to detect interactions among biological entities, but they are frequently insensitive to weak and transient interactions [26].

Nowadays, taking advantage of large-scale omics data and annotations in databases, bioinformatics-based high-throughput analyses are widely used to infer biological knowledge by applying holistic approaches from human health to agricultural sciences [27]. In this sense, network-based strategies have great potential since networks can be combined with enrichment, clustering, data-guided prioritization, and other approaches for biomarker discovery and confirmation [28]. These holistic approaches resolve some problems associated with the traditional methods based on the comparison of group pairs, such as differentially expressed gene (DEG) analysis, and thus have more advantages than traditional methods, being able to comprehensively explore the associations between genes and target traits [29]. Among a variety of methods applied to infer networks, weighted gene co-expression network analysis (WGCNA) has shown its potential in developing an understanding of the relationships among genes and in focusing on specific modules of co-expressed genes and key genes correlated with specific phenotypes and, thus, have been used to select targets or to characterize traits [12,30,31]. Additionally, expression data can be analyzed by traditional methods based on pairwise comparisons to identify the statistically significant variations of the expression patterns to profile traits, uncover targets, or focus on specific biomarkers in plants [32,33]. In this perspective, the identification of DEGs is a strong and relatively simple analysis whose application for validating key genes derived via WGCNA is a robust practice that has become very common [34,35].

Thus, gaining knowledge from available RNA-seq data and based on co-expression analysis, the present study aimed to deepen the understanding of *A. thaliana* and *A. halleri* root responses to Cd stress by performing an in silico identification and comparison among key pathways and genes underlying the phytostabilization and hyperaccumulation traits. Specifically, RNA-seq data were used to derive highly correlated modules of co-expressed genes associated with Cd responses. By analyzing the interaction network between the co-expressed genes of the two *Arabidopsis* species (*halleri* and *thaliana*) and the Cd-related genes, the ultimate goal was to identify the hub genes and preliminarily explain the reasons for the different abilities of the species in terms of Cd responses. Moreover, the detection of DEGs under Cd stress was used to confirm the key genes as reliable markers, and data retrieved from databases, including literature, were used for additional confirmation. Key functions and genes can be considered as further in wet strategies or to be furtherly explored by in silico techniques to strengthen phytoremediation knowledge and related technologies.

## 2. Results

### 2.1. Identification of Functionally Active Genes of A. thaliana Root

Various studies related to RNA-seq analysis were selected by a literature search to identify genes expressed in the root of *A. thaliana*. Other genes were retrieved by exploiting the functional annotation by GO terms as reported in the TAIR database. This led to a comprehensive list of 11,897 functionally active genes in *A. thaliana* root (Figure 1), of which only 3502 (29.44% of the total) were present in two or more datasets.

### 2.2. Co-Expression Network of A. thaliana Root

The application of WGCNA was performed to group the functionally active genes of *A. thaliana* in the root according to their expression pattern and to identify modules. Specifically, the raw counts of the reads of the 11,897 genes of *A. thaliana* considered as functionally active in the root were generated from RNA-seq data and submitted to WGCNA. Prior to the module identification and analysis, all parameters of WGCNA were screened. Specifically, a soft threshold power of β equal to 18 was set to calculate the adjacency matrix on a scale-free topology with R^2^ = 0.91 (Figure 2).

The adjacency matrix was first converted into TOM and then into 1-TOM to group genes in modules according to their expression pattern. Modules were identified by using a clustering height cutoff of 0.25 and merging groups of genes with similar expression patterns and of a minimum size of 30 (Figure 3A). A total of 11,897 genes were grouped according to WGCNA into 20 modules (Figure 3B) with a module size ranging from 30 to 3781 genes. The module eigengenes (MEs), used as representatives of the module’s gene expression patterns, were studied to identify the existence of modules with similar expression patterns. Specifically, four clusters emerged. The first cluster comprised MElightpink4, MEcoral1, and MEnavajowhite2. The second cluster comprised MEdarkorange2, MElightblue4, MEindianred1, MEdeeppink1, and MEorange4. The third cluster comprised MEblue3, MEroyalblue3, MEthistle2, MEantiquehwite1, MEbrown4, and MEpalevioletred1. The fourth cluster comprised MEantiquewhite2, MEcoral, MEfirebrick3, MEtan3, MEdarkseagreen1, and MEmediumpurple4.

### 2.3. Identification of Modules Associated with Cadmium in A. thaliana and A. halleri

Module eigengenes (MEs), representative of the expression pattern of each group of co-expressed genes (module), were used to identify modules with the most significant variation of the expression pattern between treatment (Cd stress) and the control. This approach was applied both to *A. thaliana* root modules and also to the *A. halleri* root modules identified by Hassan and colleagues [12]. The results of this analysis allowed the selection of the lightblue4 module (corr = 0.87; *p*-value = 0.023) and the antiquewhite1 module (corr = −0.88; *p*-value = 0.01789) for *A. thaliana* root and of the brown module (corr = 0.938; *p*-value = 1.3 × 10^−11^) and the plum1 module (corr = −0.924; *p*-value = 1.33 × 10^−10^) for *A. halleri* (Figure 4). Thus, these modules were deemed as being significantly associated with Cd stress.

### 2.4. Functional Enrichment and Pathway Analysis of Cd-Associated Modules

The genes from selected modules of *A. thaliana* and of *A. halleri* were submitted to a functional enrichment analysis to identify overrepresented biological processes (BPs), molecular functions (MFs), and KEGG pathways linked to Cd stress response.

The top 3 BPs, MFs, and KEGG pathways significantly enriched in each module (*p*-value < 0.05) were identified (Figure 5), and the related genes were considered together with inferred interactions based on co-expression.

In the Cd-associated modules of both *A. thaliana* and *A. halleri*, the most enriched BPs rely on the response of the cell to various inputs, specifically are “response to stimulus” (GO:0050896) and “response to chemical” (GO:0042221) (Figure 5A, light blue and brown bars). Regarding MFs, “catalytic activity” (GO:0003824) and “ion binding” (GO:0043167) were the most overrepresented in *A. thaliana* (Figure 5B, light blue and brown bars), while in *A. halleri* the most overrepresented MFs were “catalytic activity” (GO:0003824) and “transferase activity” (GO:0016740).

The KEGG pathways found as being the most overrepresented in *A. thaliana* were “metabolic pathways” (KEGG:01100) and “photosynthesis” (KEGG:00195), while they were “Glycerophospholipid metabolism” (KEGG:00564) and “Glucosinolate biosynthesis” (KEGG:00966) in *A. halleri* (Figure 5C, light blue and brown bars). In absolute, the most overrepresented BP in *A. thaliana* was “response to stimulus” (GO:0050896; *p* < 3.12 × 10^−43^; −log(*p*) = 42.50), and for MFs it was “catalytic activity” (GO:0003824; *p* < 1.3 × 10^−18^; −log(*p*) = 17.89) and for KEGG pathways was “metabolic pathways” (KEGG:01100; *p* < 2.4 × 10^−9^; −log(*p*) = 8.60). Analogously, in the case of *A. halleri*, the most overrepresented BP and MF were, respectively, “response to stimulus” (GO: GO:0050896; *p* < 3.04 × 10^−29^; −log(*p*) = 28.51) and “catalytic activity” (GO:0003824; *p* < 2 × 10^−18^; −log(*p*) = 17.70), while the most overrepresented KEGG pathway was “Glycerophospholipid metabolism” (KEGG:00564; *p* < 0.000172; −log(*p*) = 3.76) (Appendix A).

### 2.5. Identification of Hub Genes Associated with Cd and Module Visualization and Verification

The genes from the selected modules were screened for their reciprocal expression-based relationships by using the cytoHubba plugin of Cytoscape (see methods). The top 30 hub genes with the highest Maximal Clique Centrality (MCC) (Appendix A) were selected and used to visualize the module they belong to (graphs in Figure 6). The genes associated with the highest values of MCC were deemed as hub genes.

Differentially expressed genes were identified to confirm and validate WGCNA results (Appendix A). A total of 3040 genes, of which were 55.43% upregulated and 44.57% downregulated, were found in the root of *A. halleri* under Cd stress, while 5507 genes, of which 47.40% were upregulated and 52.60% downregulated, were found in *A. thaliana* root under Cd stress. Volcano plots of these identifications together with their functional enrichment analysis are presented in Appendix A. The three functions, processes, and pathways mostly significantly overrepresented in DEGs (Table 1) included upregulated structural functions and processes of responses to oxidative stress and downregulated catalytic activities and protein/macromolecule modifications in *A. halleri*.

In the case of *A. thaliana*, instead, upregulated DEGs are associated with catalytic/binding functions and processes of cell wall reorganization, while downregulated DEGs are associated with small molecule binding and response to diverse stimuli.

By comparing the hub genes identified in the modules of interest and the DEGs, we found that 7 out of 30 hub genes of the brown module (Figure 6a) and all the hub genes of the plum1 module (Figure 6b) intersected with the DEGs in the case of *A. halleri*. Similarly, the hub genes from the lightblue4 module (Figure 6c) were all intersecting with the DEGs, while 27 out of 30 hub genes of the antiquewhite1 module (Figure 6d) were intersecting with DEGs in *A. thaliana* (Appendix A).

In the case of the lightblue4 module of *A. thaliana*, the top three protein-coding hub genes with the highest MCC score were pentatricopeptide repeat (PPR) superfamily protein (AT1G71210), HIT-type zinc finger family protein (AT5G63830), and metal-beta lactamase family protein (AT4G33540). The hub genes of the antiquewhite1 module of *A. thaliana*, instead, codified for proline-rich protein 1 (PRP1, AT1G54970), transmembrane protein (AT2G28410), and O-glycosyl hydrolases family 17 protein (AT3G07320). The hub genes of the brown module of *A. halleri* codified for a transcription factor of the NAC (No apical meristem (NAM), Arabidopsis transcription activation factor (ATAF), Cup-shaped cotyledon (CUC)) superfamily (AT3G10500), an O-fucosyltransferase family protein (AT3G02250), and for a trichohyalin (AT3G15550). Instead, the hub genes of the plum1 module codified for the hypothetical protein O-Glycosyl hydrolases family 17 protein (XP_003322588), for the signal recognition particle, SRP54 subunit protein (AT5G49500), and for the ubiquitin-conjugating enzyme 36 (NP_849678). These top three hub genes were submitted to ThaleMine [41] to retrieve knowledge about their possible involvement in Cd or any other metal stress or in general stress responses. Specifically, all publications present on each ThaleMine gene page were screened about their content to relate genes to Cd/metal/general stress responses (Appendix A).

Overall, the verification obtained by intersecting the genes is a sign of the reliability of the hub genes and, in general, of these results.

## 3. Discussion

Heavy-metal-polluted soils are, nowadays, spread all over the world, representing a vital issue for agriculture and related productions, food safety, and toxic effects on plants, animals, and humans [42]. Thus, the mitigation of heavy metal presence in contaminated soils is required to prevent contamination and subsequent health problems. Different possible strategies adopted to address this task rely on physico-chemical techniques, which are costly, hard-to-apply, ineffective, and, additionally, they frequently introduce secondary pollutants and irreversibly alter soil properties [43]. Another strategy is based on the usage of plants to remediate heavy-metal-polluted soils, by phytoremediation, and is particularly promising for its effectiveness, simplicity of application, and public acceptance [44].

During the past decades, hundreds of hyperaccumulator plants have been identified and a wide variety of in-wet experiments have been attempted to achieve a careful description of the biology underlying this trait [45,46]. Despite this, the hyperaccumulation trait still needs to be dissected and, in this sense, bioinformatics may provide tools for the comprehensive analysis and modeling of the related processes, as already applied for various purposes in plant biology studies [4,47,48,49,50].

Thus, this work aimed to exploit already available high-throughput data to obtain a comprehensive and accurate mapping of protein–protein interactions from which novel biological knowledge can be derived. Specifically, we studied the phytostabilization and hyperaccumulation traits of the roots of *A. halleri*, together with an analysis and comparison with the phytostabilization ability of the roots of *A. thaliana*, characterizing their responses to Cd stress. Weighted gene co-expression network analysis was applied to RNA-seq data of *A. thaliana* roots grown in the presence of Cd and in control conditions, allowing the identification of 20 co-expressed groups of genes (modules). Specifically, to identify the most comprehensive list of *A. thaliana* genes expressed in the roots, several expression datasets were considered and compared, showing only a partial overlap. On the base of our observations, only a small percentage of identified genes overlapped among two or more experiments of RNA-seq, while a large amount of genes were identified uniquely in each single RNA-seq experiment. RNA expression is strongly influenced by time, conditions, and possible perturbations of the organism under analysis, and this largely affects RNA-seq reproducibility and gene identification [51,52]. Thus, via meta-analysis, it is possible to integrate data from multiple independent studies, improving reproducibility and enabling obtaining all-encompassing datasets [53].

The modules from *A. thaliana* were considered together with the 14 modules of co-expressed genes identified by Hassan et al. [12] from RNA-seq of *A. halleri* roots grown under Cd stress and in control conditions. All module eigengenes, considered representative of the expression pattern of the genes within each module, were used to identify the modules showing the most statistically significant difference of expression between the Cd-treatment versus the control. By WGCNA, each gene was placed in its functional context which, in the case of the modules of interest, has been enriched by GO terms and KEGG pathway to identify possible mechanisms commonly involved in the phytostabilization process or possible differences able to evidence mechanisms of the hyperaccumulation trait peculiar to *A. halleri*. This analysis indicated that in *A. halleri* roots the genes with the strongest significant variation (both positive and negative) among Cd stress and control are, in any case, associated with responses to chemical and generic stimuli. This agrees with the evidence that *A. halleri* can perceive and respond appropriately to different environmental stimuli, modulating and integrating multiple stimulus information [54] and giving a specific response to a peculiar heavy metal [55]. Specifically, Cd induces oxidative stress due to the impairment between photosynthetic and respiration activities [56], which has also been described in *A. halleri* [12]. Moreover, macromolecule modifications, such as glycosylation, are known mechanisms to be strongly active in plants under stress conditions in which the redox status acts as a central hub connecting enzymatic reactive oxygen species (ROS) scavenging to biosynthesis and signaling molecules [57]. In this perspective, our results suggest that a similar strategy, based on a careful integration of stimuli in which oxidative stress has a pivotal role, can also be possibly used to cope with the stress due to Cd presence.

In the case of *A. thaliana*, also, the genes with an expression significantly higher under Cd stress are enriched in functions associated with stimuli responses, but the genes with a significantly lower expression under Cd stress are associated with root development. Notably, it has already been reported that heavy metals have deleterious effects on plants such as morphological abnormalities, reduced dry weight, decreased germination, and reduced elongation of both roots and shoots [58]. Specifically, Cd induces DNA damage and cell cycle arrest in *A. thaliana* root tips [59] and it is responsible for a deep root architecture rewiring, negatively affecting primary root formation and somehow enhancing the formation of lateral ones [60]. In particular, the excess accumulation of nitric oxide (NO) in *A. thaliana* under Cd exposure leads to the harmful effect of inhibiting primary root growth, reducing the activation of auxin signaling [61]. Thus, our findings emphasize the importance of *A. thaliana* architectural plasticity in response to Cd and suggest that changes in root architecture may allow plants to effectively exploit soil. At the cellular level in *A. thaliana*, cell wall remodeling is used to prevent Cd from entering into the cells [62] and to face other types of stresses such as spaceflight adaptation in the absence of gravity [63]. To further investigate the pivotal functions of the root under Cd stress, WGCNA results were explored to detect the hub genes as those, within the modules of interest, with the highest MCC score [64]. Indeed, abnormal responses of plants to stress rely on the elimination of crucial players from the stress-related networks [65]. This is because plants respond to specific stresses by modulating complex interconnected pathways in which some genes play crucial roles both in generalized and specific responses to stress [54,55]. In this sense, after hub identification, we detected genes with a significant differential expression under Cd stress by DESeq2 package to confirm hub gene predictions [34,35]. The intersection between DEGs and hub genes from WGCNA, together with enrichment analysis, verified the reliability of the results.

Later, data mining of the top three hub genes of each module of interest was also supported by the usage of ThaleMine [41] to retrieve data about their involvement in the response of the plants to Cd stress or other types of stresses. The hub gene AT1G71210, identified in the lightblue4 module of *A. thaliana*, codified for a pentatricopeptide repeat (PPR) superfamily protein. This gene in soybean is regulated under heat stress conditions [66], while in *Arabidopsis arenosa*, a species closely related to *A. halleri* and able to hyperaccumulate zinc and Cd, is part of a group of five genes that are under a convergent selective sweep between the metallicolous and non-metallicolous populations [67]. However, despite this, no evidence supports any description of its role in *A. thaliana*, including the information retrieved by ThaleMine even if the pentatricopeptide repeat proteins are described as strongly involved in abiotic stress responses [68]. The orthologs in rice of the gene AT5G63830, also identified as one of the top three hub genes in the Cd-associated lightblue4 module of *A. thaliana* and codifying for HIT-type zinc finger family protein, is involved in mineral homeostasis after water deficit by an auxin-mediated process [69]. Auxin signaling is known to be involved in responses associated with heavy metals and metalloids specifically by its crosstalk with ROS in *Arabidopsis* plants [70]. However, a detailed description of the possible involvement of this gene in a process by which auxin modulates response to Cd at physiological and molecular levels is still missing. The gene AT4G33540 (metal-beta-lactamase family protein), within the top three hub genes of the lightblue4 module of *A. thaliana*, has been demonstrated to be strongly regulated in mutants lacking ARR6, a member of the Arabidopsis Response Regulators (ARRs), which is involved in the modulation of some disease resistance responses [71]. The data retrieved by ThaleMine evidenced that this gene is involved in *A. thaliana* responses to pathogens, it is inducible by arsenic, and it is modulated during early responses to ethylene. Exploring ThaleMine suggested literature also showed that AT4G33540 interacts with atToc132, a receptor needed for the post-translational import of nucleus-encoded chloroplast pre-proteins. Specifically, atToc132 binds both photosynthetic and non-photosynthetic pre-proteins as an adaptation of an optimum balance of chloroplast proteins to be maintained at all stages of development and plastid biogenesis, as well as under biotic and abiotic stress conditions [72]. These suggest the involvement of the AT4G33540 gene in stress responses in *A. thaliana*, but more data are needed to confirm.

The top three hub genes of the antiquewhite1 module of *A. thaliana*, which is negatively associated with Cd presence, were searched in literature and ThaleMine. In particular, the gene AT1G54970, which codifies for Proline-rich protein 1 (PRP1), is a gene expressed only in the root of *A. thaliana* and is more likely involved in epidermal cell differentiation [73,74] as confirmed by literature reported in ThaleMine. Additionally, still analyzing ThaleMine retrieved data, this gene seems to be involved in multiple hormonal regulations of *Arabidopsis* development and specifically in root growth and root hair development. Thus, it is not surprising that we can predict this gene as a hub involved in *A. thaliana* root response to Cd stress since root remodeling and growth inhibition have been widely described as a consequence of heavy metal stress [58,59]. Another top hub gene from the antiquewhite1 module is AT2G28410, which is predicted as a transmembrane protein, but with no function annotated so far. This gene, from ThaleMine data mining, is involved in responses to pathogens and has been described as an abscisic acid (ABA)-repressed gene in guard cells [75]. Guard cells are affected by Cd which uses calcium channels to permeate them [62]. Other signs of signal transduction involvement arrive from the prediction of this gene as a glycosylphosphatidylinositol (GPI)-anchored protein (GPI-APs), which are known to be involved in many developmental progresses in *Arabidopsis* [76]. Recently, AT2G28410 has been predicted as part of the “glucosinolate biosynthesis” pathway [77]. Under Cd exposure *A. thaliana* roots tend to accumulate glucosinolates, which are known to act in anti-herbivore and anti-aphid defense [12,78]. Thus, even if there is no evidence supporting a defined role of AT2G28410 in the *A. thaliana* root responding to Cd stress, several data support its predicted involvement in the process. The gene AT3G07320, identified as the top hub gene in the antiquewhite1 module of *A. thaliana* and codifying for the O-glycosyl hydrolases family 17 protein, from ThaleMine research emerges as being involved in plants responding to pathogens and during xylem differentiation. Moreover, it has been demonstrated to be able to interact with DNA and to be a cell wall component in *A. thaliana* [79,80]. In fact, as reviewed by Loix and colleagues [62] and already described, plants have a variety of tools to prevent Cd from entering cells, including a deep remodeling of the cell wall. The cell wall is also a key trait of the spaceflight adaptation of *A. thaliana* under gravitropism stress in which the gene AT3G07320 plays a role together with other genes such as AT1G63440, AT1G71050, and AT1G12950 codifying respectively for HMA5 heavy metal ATPase 5, for HIPP20 heavy metal transport/detoxification superfamily protein, and for RSH2 root hair specific 2, which are generally involved in heavy metal stress responses and related root remodeling [63]. Additionally, in this case, even if there is no direct evidence, the involvement of AT3G07320 in metal stress responses and specifically to Cd can be investigated.

In the case of *A. halleri*, the top hub genes from modules of interest were also identified and searched through the ThaleMine database. In the case of AT3G10500, one of the top three hub genes of the brown module of *A. halleri* is positively associated with Cd, and various publications mentioned it as being involved in various types of stresses including cold, pathogens, drought, and heavy metals. Specifically, AT3G10500 is one of the most significant genes expressed differentially in the root under cesium stress [81]. Intriguingly, this gene codifies for a NAC[No apical meristem (NAM), Arabidopsis transcription activation factor (ATAF), Cup-shaped cotyledon (CUC)], which is a member of a superfamily of transcription factors known to be proteolytically activated and translocated to the nucleus during responses to abiotic stresses [82]. Crucial signaling components such as calcium signaling, hormone signaling, and mitogen-activated protein kinase (MAPK) signaling are based on the accumulation of ROS due to the presence of abiotic stress [83]. However, despite this, the proper activation of AT3G10500 in *A. halleri* must still be investigated. The results of data mining from the search in ThaleMine of the gene AT3G02250, codifying for a protein of the family of the o-fucosyltransferase, indicate its involvement in responses to pathogens. A further literature search showed that mechanisms of glycosylation are generally used by plants to face growing conditions favoring oxidative stress [57]. Cadmium is known to impair photosynthetic and respiration activities inducing oxidative stress [56] and also in *A. halleri* [12], but the mechanisms underlying this in which AT3G02250 is involved still need comprehension. The data retrieved from ThaleMine about the gene AT3G15550, another of the top three hub genes from the brown module, which codify for trichohyalin, are affected by poor knowledge. However, some indication of a possible involvement of it in stress responses is related to a trichohyalin-like gene (AT4G27980) which is upregulated under drought stress conditions in *A. thaliana* [84]. Two out of three top three hub genes identified in the plum1 module gave no results in ThaleMine as they are predicted/hypothetical genes. Instead, the data retrieved about the last top hub gene of the plum1 module of *A. halleri*, showed that the gene AT5G49500 (SRP54 subunit protein) has a possible involvement in plant hormone signaling which is implicated in stress responses [62,75].

Overall, the WGCNA, with comprehensive RNA-seq datasets, can provide key functions to unveil genes involved in root responses of *A. thaliana* and of *A. halleri* to Cd stress. Additionally, these key genes are good candidates that may deserve as targets for further investigations, also considering inter-kingdom network-based approaches, which specifically are used to characterize the association between the transcriptional dynamics of co-expression data and the identification of other components of the rhizosphere, such as bacteria/fungi, widely used in assisted phytoremediation strategies [85,86,87].

## 4. Materials and Methods

### 4.1. Identification of A. thaliana Genes Functionally Active in the Root

A review by literature search was performed to provide recent comprehensive studies on RNA-seq analysis of the transcriptomes of *A. thaliana* root. This was performed in January 2022 using queries containing the keywords “*Arabidopsis thaliana*”, “RNA-seq”, and “root” on Pubmed (https://pubmed.ncbi.nlm.nih.gov/, accessed on 10 January 2022) and Google Scholar (https://scholar.google.com/, accessed on 17 January 2022) databases. Of all the literature documents resulting from the searches, only the original articles published between 2020 and 2021 were screened and retained if describing lists of genes identified as expressed, and thus functionally active, in the organ of the plant.

Gene Ontology terms (GO terms) are used as attributes to identify biological processes, molecular functions, or cellular localizations associated with a gene product. The Arabidopsis Information Resource (TAIR) database (https://www.arabidopsis.org/, accessed on 25 January 2022) was exploited to retrieve genes annotated with GO terms including the word “root”. Thus, all *A. thaliana* genes annotated with GO terms including “root” were used to compile a list of genes functionally associated with the root. These were compared and integrated with the lists of the genes identified as functionally active from transcriptomic studies by using InteractiVenn [88]. 

### 4.2. Data Source and Analysis of Expression Data of A. thaliana Root

The web-available tool RaNA-seq [89] was applied to RNA-seq FASTQ data of *A. thaliana* to derive raw counts related only to identified genes active in the root. In detail, RNA-seq data were obtained from Pacenza and colleagues [90] that analyzed transcripts from mutant (*ddc*) and wild-type (Columbia ecotype) *A. thaliana* plants grown in the presence of a medium supplemented with CdCl_2_ for a final concentration of Cd of 25 μM or 50 μM, and in control conditions. Sequencing data were downloaded at the Sequence Read Archive (SRA, https://www.ncbi.nlm.nih.gov/sra, accessed on 1 September 2022) of the National Center for Biotechnology Information within the BioProject identification number PRJNA641242. The row counts of genes identified as functionally active in the root of the plant were considered organ-specific expression datasets. The free accessible R package WGCNA [30] was used to identify co-expression patterns and construct a signed weighted gene co-expression network. Specifically, after removing low-count genes (<50 reads) and normalizing RNA-seq counts by the “*vst*” function from thhhe DESeq2 package [91], data were transformed by removing the dependence of variance on the mean. A similarity matrix was obtained to calculate the correlation of Pearson among expression data of each gene pair. This was converted into an adjacency matrix by using a soft threshold power β selected to encourage strong correlations (discouraging weak ones) for creating a scale-free network. The adjacency matrix which was then converted into a topological overlap matrix (TOM) defines the strength of the association between genes. The dissimilarity TOM was calculated and used by applying a DynamicTreeCut algorithm (version 1.63-1) to group genes with similar expression patterns and to identify the correct number of modules. A module was considered only if containing a minimum number of genes equal to 30. The module merging threshold was set to 0.25. Co-expression similarity among modules was determined by the flashClust function and, as representative of the expression pattern of the module, module eigengenes (MEs) were determined by calculating a weighted average of the expression patterns of all genes within a module. A heatmap was used to visualize the correlation among MEs.

### 4.3. Identification of the Module of Interest

Co-expression modules significantly associated with Cd treatment (*p*-value < 0.05) were identified by the rcorr package in R [92] with associated Pearson Correlation value and significance showed by heatmap. Modules most significantly associated with Cd treatment were considered as key modules and considered for the following analysis. This was performed on *A. thaliana* and *A. halleri* by correlating their MEs with the presence (or absence) of Cd in the sample.

### 4.4. Functional and Pathway Enrichment Analysis

The Gene Ontology (GO; http://www.geneontology.org/, accessed on 1 November 2022) terms for molecular functions (MFs) and biological processes (BPs) together with the Kyoto Encyclopedia of Genes and Genomes (KEGG) pathways were searched among genes sharing a similar expression pattern (modules). Overrepresented MFs, BPs, and KEGG pathways (adjusted *p*-value < 0.05) were identified through g:GOSt, at the g:Profiler web server for functional enrichment analysis [93], and the top 3 ranking GO terms or KEGG pathways were analyzed and compared. Similarly, overrepresented MFs, BPs, and KEGG pathways were identified also among genes found to be differentially expressed under Cd stress.

### 4.5. Hub Gene Identification

The Maximal Clique Centrality (MCC) algorithm, available with the cytoHubba (v. 0.1) [64] plugin of Cytoscape (v. 3.8.2) [94], was used to rank nodes according to their relevance to the network topology. Top MCC-scoring genes were deemed as hub genes as they represent the genes most strongly interconnected with other genes representing genes with pivotal functions and positions in the network [95,96]. The module of interest from both *A. halleri* and *A. thaliana* were submitted to cytoHubba.

### 4.6. Identification of Genes Differentially Expressed under Cd Stress

For detecting how Cd presence can affect gene expression in both *A. thaliana* and *A. halleri* roots, counts related to gene expression were first normalized and then, imputing by means of control data to be coherent with 3 samples [97], the differentially expressed genes (DEGs) were screened using DESeq2 [88] package in R using padj < 0.1 and log_2_ Fold Change > 0.5. DEGs visualization was performed by a plot through Enhancedvolcano [98] and their intersection with WGCNA hub genes was presented as a Venn diagram obtained by Venny [99].

## 5. Conclusions

The pipeline applied in this study is based on the application of weighted gene co-expression analysis (WGCNA) and on the detection of the differentially expressed genes to identify traits underlying Cd homeostasis in the root organ of the model *A. thaliana* and of the hyperaccumulator *A. halleri*. The results obtained from this study permitted us to focus on groups of genes specifically associated with the presence of Cd that here are shown to be pivotally involved in the related responses. Under Cd stress, in detail, the genes of the roots of a hyperaccumulator such as *A. halleri*, mostly associated with chemical and generic stimuli responses, are finely regulated (up- and down-regulated) by a specific pattern. In the roots of *A. thaliana*, which phytostabilize, the genes associated with the same responses (chemical and generic stimuli responses) were only upregulated, while downregulated genes were associated with root development to drastically modify the organ at the physiological and cellular levels. Moreover, the hubs among these genes were confirmed by overlapping DEGs and the top three of them were furtherly confirmed by the analysis of external sources of knowledge from the ThaleMine database and literature. This validated them as being involved in Cd stress and elevated these genes as robust candidates for further investigations and approaches and to drive future research in the field.

## Figures and Tables

**Figure 1 plants-12-01793-f001:**
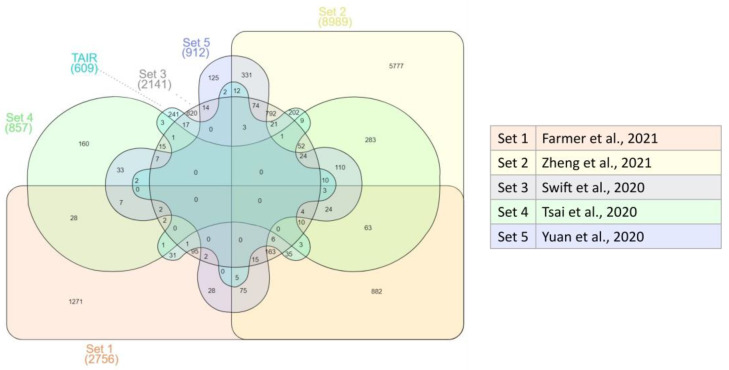
The functionally active genes of *A. thaliana* root: the Venn diagram shows the overlap among the dataset of genes expressed in the root of *A. thaliana* (set1, set2, set3, set4, and set5) or functionally annotated by GO terms including “root” as reported in the highly curated database TAIR (TAIR). The table shows the references to the expression data [36,37,38,39,40].

**Figure 2 plants-12-01793-f002:**
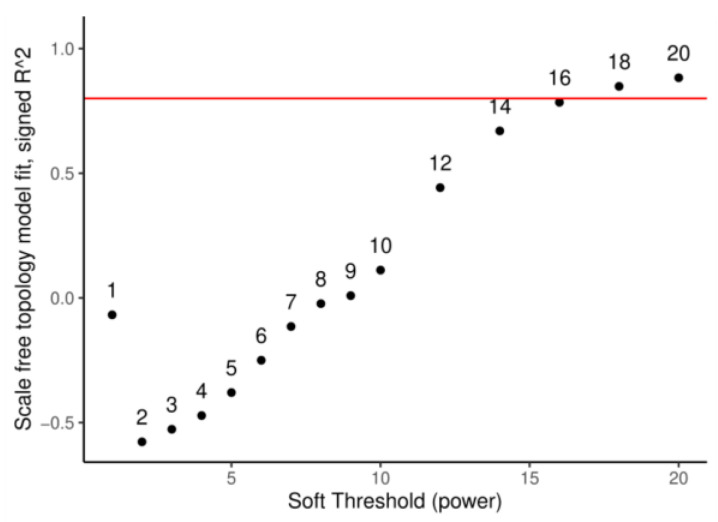
Topology of the co-expression network of *A. thaliana* root: topological analysis of the co-expression network by thresholding powers (numbers in the plot). The red line marks the value of the scale-free fit index (0.8).

**Figure 3 plants-12-01793-f003:**
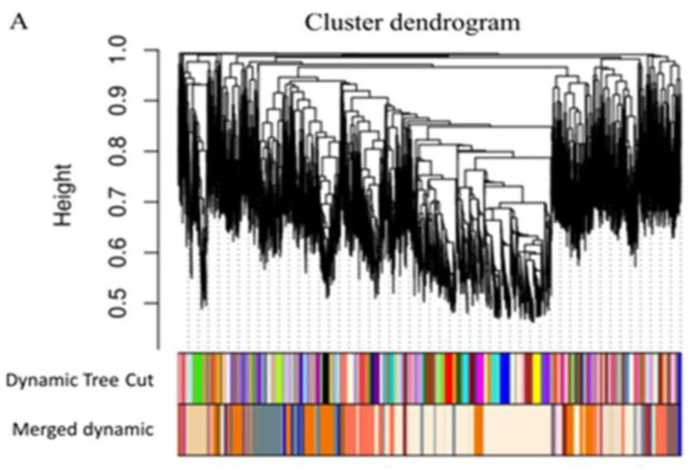
Weighted gene co-expression network analysis of *A. thaliana* genes active in the root: dendrograms showing the hierarchical clustering and grouping of genes (**A**) and of modules (**B**). Initial and after-merging modules are shown by color bars (**A**). Each module is represented by its module eigengene (ME) and the threshold cutoff applied to merge modules is shown by red line (**B**). The heatmap (**C**) correlates MEs: red for high adjacency and blue for low ones. Lateral color bars identify modules.

**Figure 4 plants-12-01793-f004:**
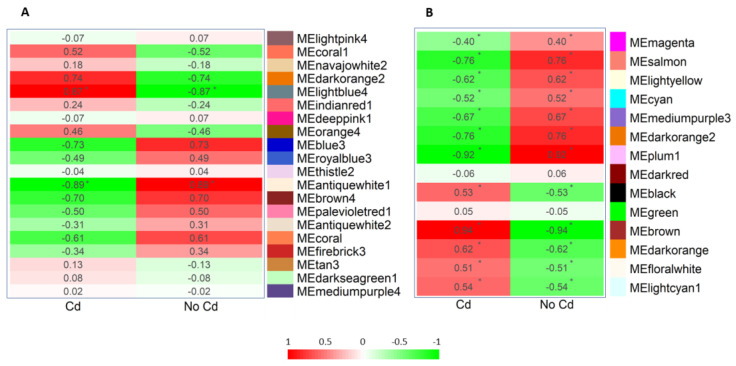
Module–cadmium association: correlation among module eigengenes (MEs) and the pattern of Cd treatment presented by heatmap for *A. thaliana* modules (**A**) and *A. halleri* modules (**B**). Rows represent modules, while columns are for Cd-treatment (Cd) or control (No Cd). Each cell color represents the corresponding Pearson correlation coefficient computed between a module eigengene and the Cd-related pattern. Red-to-green shows positive-to-negative values. Asterisks show significant correlations (*p*-value < 0.05).

**Figure 5 plants-12-01793-f005:**
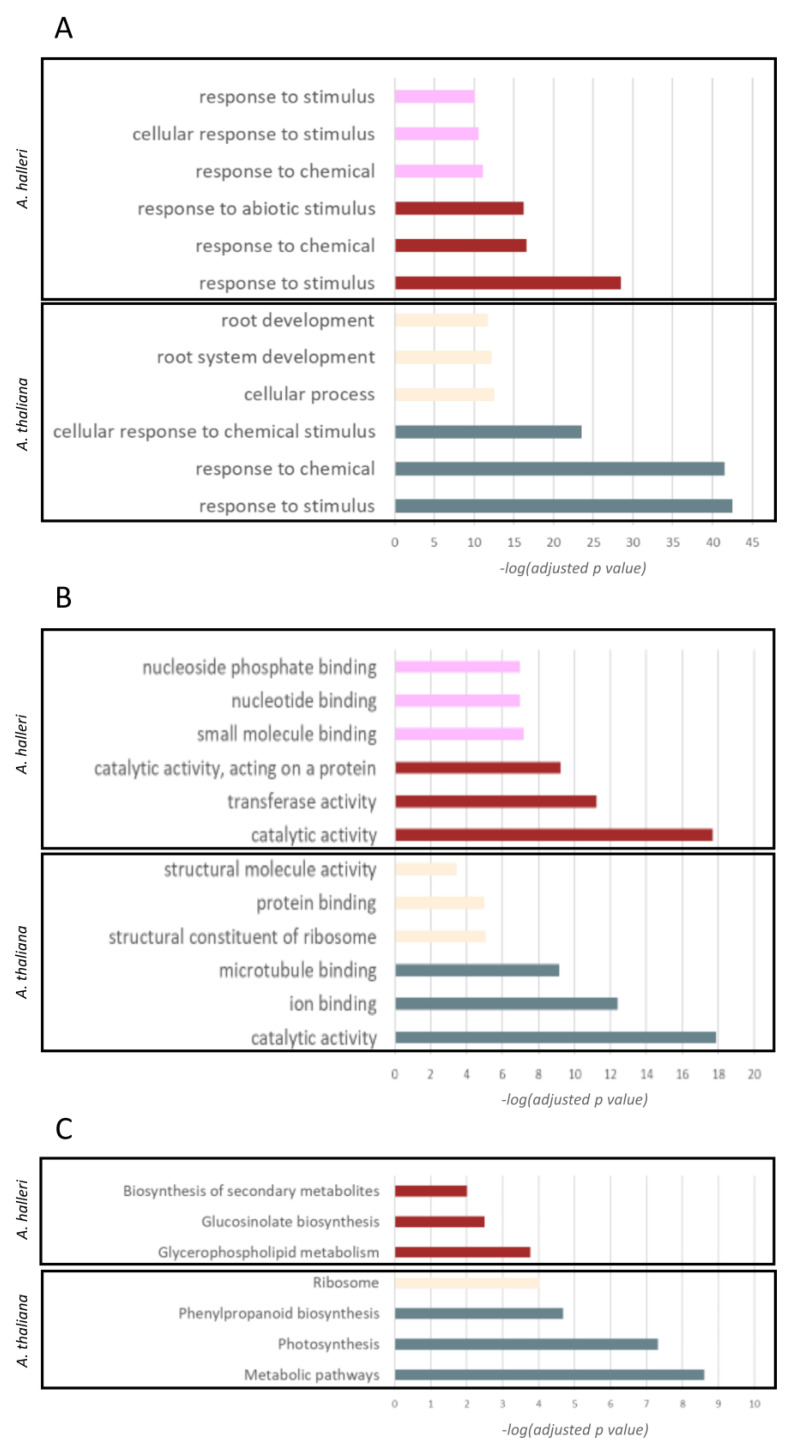
Gene ontologies and KEGG pathway enrichment: bars represent the top three overrepresented biological processes (**A**), molecular functions (**B**), or KEGG pathways (**C**) identified for the genes belonging to selected Cd-associated modules in both *A. thaliana* and *A. halleri*. Bar colors trace module colors and the x-axis represents −log_10_ (adjusted *p* value).

**Figure 6 plants-12-01793-f006:**
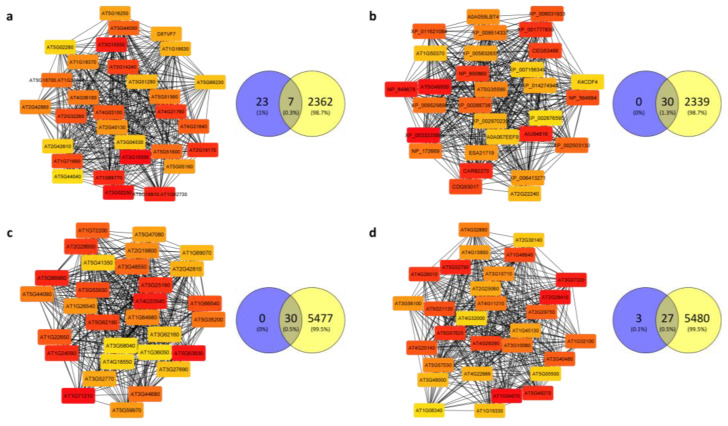
Visualization of Cd-associated modules and overlap of hub genes with differentially expressed genes: the top 30 genes with the highest MCC values were deemed as hub genes and are visualized from the brown module (**a**) and plum1 module (**b**) of *A. halleri* and from the lightblue4 module (**c**) and the antiquewhite1 module (**d**) of *A. thaliana* together with a Venn diagram reporting the hub genes overlapping with the DEGs. Nodes represent the hub genes as rectangles and their color is red-to-yellow from the highest MCC value onwards. Edges represent co-expression detected by WGCNA. In the Venn diagrams blue for hub genes and yellow for DEGs, their overlap is in the intersection.

**Table 1 plants-12-01793-t001:** The top three MFs, BPs, and KEGG pathways overrepresented among upregulated and downregulated DEGs of *A. halleri* and *A. thaliana*.

		Source	Term Name	Term Id	Padj
*A. halleri*	Downregulated	GO:MF	catalytic activity, acting on a protein	GO:0140096	7.44
		GO:MF	catalytic activity	GO:0003824	5.83
		GO:MF	purine ribonucleoside triphosphate binding	GO:0035639	5.66
		GO:BP	protein modification process	GO:0036211	10.86
		GO:BP	macromolecule modification	GO:0043412	9.91
		GO:BP	phosphorus metabolic process	GO:0006793	8.58
		-			
		-			
		-			
	Upregulated	GO:MF	structural molecule activity	GO:0005198	21.71
		GO:MF	structural constituent of ribosome	GO:0003735	15.21
		GO:MF	ribonucleoside triphosphate phosphatase activity	GO:0017111	14.45
		GO:BP	cellular response to hypoxia	GO:0071456	25.03
		GO:BP	cellular response to decreased oxygen levels	GO:0036294	24.85
		GO:BP	cellular response to oxygen levels	GO:0071453	24.85
		KEGG	Ribosome	KEGG:03010	10.08
		KEGG	Phagosome	KEGG:04145	5.28
		KEGG	Proteasome	KEGG:03050	1.61
*A. thaliana*	Downregulated	GO:MF	ion binding	GO:0043167	16.74
		GO:MF	catalytic activity	GO:0003824	13.48
		GO:MF	small molecule binding	GO:0036094	13.33
		GO:BP	response to stimulus	GO:0050896	60.66
		GO:BP	response to chemical	GO:0042221	46.39
		GO:BP	response to stress	GO:0006950	42.47
		KEGG	Valine, leucine and isoleucine degradation	KEGG:00280	5.00
		KEGG	Fatty acid degradation	KEGG:00071	1.98
		KEGG	Circadian rhythm-plant	KEGG:04712	1.86
	Upregulated	GO:MF	catalytic activity	GO:0003824	14.49
		GO:MF	protein domain specific binding	GO:0019904	14.19
		GO:MF	microtubule binding	GO:0008017	11.10
		GO:BP	cell wall organization or biogenesis	GO:0071554	37.77
		GO:BP	cell wall organization	GO:0071555	31.54
		GO:BP	external encapsulating structure organization	GO:0045229	30.54
		KEGG	Metabolic pathways	KEGG:01100	22.12
		KEGG	Photosynthesis	KEGG:00195	14.62
		KEGG	Phenylpropanoid biosynthesis	KEGG:00940	8.85

## Data Availability

All datasets on which the conclusion of the paper relies are available in the paper itself or included in the Appendix A.

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
