# Peer review of "Key Pathways and Genes of Arabidopsis thaliana and Arabidopsis halleri Roots under Cadmium Stress Responses: Differences and Similarities"

_plants, 2023, doi:10.3390/plants12091793_

Round 1

Reviewer 1 Report

The manuscript lacks experimental data obtained by the authors. It is devoted to a comparative analysis of gene expression in the roots of the hyperaccumulator Arabidopsis halleri and the closely related excluder Arabidopsis thaliana under the action of Cd based on the data from several sources. I am not an expert in the field of WGCNA analysis, but having read the manuscript, I have a fundamental question. Obviously, the accumulation of Cd in plant roots depends not only on its concentration in the medium, but also on plant growth conditions. The level of gene expression will depend on metal concentration in root cells, which is of fundamental importance, especially when comparing a hyperaccumulator, which accumulates Cd mainly in shoots, and an excluder, which accumulates Cd mainly in roots and is much less tolerant. Therefore, strictly speaking, even at the same external concentrations, the accumulation of metal in the roots of A. thaliana or A. halleri will differ under different experimental conditions, which makes it difficult to compare the data obtained by different authors. If metal concentration in the roots differs in different experiments, the degree of Cd-imposed stress will be different as well. It is obvious that the databases contain the results of experiments done by different authors, which were carried out at different Cd concentrations, plant growth conditions, and duration of incubation in the presence of the metal, which cannot but affect the expression of certain genes. The question arises: how legitimate is it to analyze the entire set of data obtained earlier by different authors, without taking into account the concentration of Cd and the growth conditions? As to the minor concerns, it is necessary to decipher the abbreviations in the text at their first mention.

Reviewer 2 Report

The manuscript is very important in field 

But need improved language editing

In part conclusion where the the effect of heavy metal on plants

This paper may be useful for you

Biosorption effect of Aspergillus niger and Penicillium chrysosporium for Cd and Pb contaminated soil and their physiological effects on Vicia faba L. Environmental Science and Pollution Research. 28(47):67608-67631.

2-Microbe-Assisted Phytoremediation of Environmental Pollutants and energy recycling in Sustainable Agriculture. Archives of Microbiology 203: 5859–5885. https://doi.org/10.1007/s00203-021-02576-

Line 349-360 please parpherase it

Round 2

Reviewer 1 Report

The authors have provided the explanation of the methods used.